# Dual-Resonator-Based (DRB) and Multiple-Resonator-Based (MRB) MEMS Sensors: A Review

**DOI:** 10.3390/mi12111361

**Published:** 2021-11-04

**Authors:** Yusi Zhu, Zhan Zhao, Zhen Fang, Lidong Du

**Affiliations:** 1State Key Laboratory of Transducer Technology, Aerospace Information Research Institute, Chinese Academy of Sciences, Beijing 100190, China; zhuyusi16@mails.ucas.ac.cn (Y.Z.); zhaozhan@mail.ie.ac.cn (Z.Z.); zfang@mail.ie.ac.cn (Z.F.); 2School of Electronic, Electrical and Communication Engineering, University of Chinese Academy of Sciences, Beijing 100049, China

**Keywords:** dual-resonator-based (DRB) MEMS sensor, multiple-resonator-based (MRB) MEMS sensor, strength-coupled-resonator-based (SCRB) sensor, wave-coupled-resonator-based (WCRB) sensor, uncoupled-resonator-based (UCRB) sensor

## Abstract

Single-resonator-based (SRB) sensors have thrived in many sensing applications. However, they cannot meet the high-sensitivity requirement of future high-end markets such as ultra-small mass sensors and ultra-low accelerometers, and are vulnerable to environmental influences. It is fortunate that the integration of dual or multiple resonators into a sensor has become an effective way to solve such issues. Studies have shown that dual-resonator-based (DRB) and multiple-resonator-based (MRB) MEMS sensors have the ability to reject environmental influences, and their sensitivity is tens or hundreds of times that of SRB sensors. Hence, it is worth understanding the state-of-the-art technology behind DRB and MRB MEMS sensors to promote their application in future high-end markets.

## 1. Introduction

In the past two decades, single-resonator-based (SRB) Micro-Electro-Mechanical-Systems (MEMS) sensors have been extensively studied for detecting small physical or chemical perturbations, such as pressure [1,2,3,4,5,6,7,8], rotational rate [9,10,11,12,13,14], acceleration [15,16,17], vibration [18,19,20], force [21,22,23], chemical vapor [24,25,26], and biological material [27,28,29,30,31,32,33,34]. Various structures, such as flexural mode resonators (FMR) [28,31,35], surface acoustic wave resonators (SAW) [36], bulk acoustic resonators (BAR) [6,12,24,25,26,37], lamb wave resonators (LWR) [8,14,38], etc., have been fabricated to enable these SRB sensors to have an excellent performance. A key attribute of these sensors is that the output signal is the variation/shift in the resonant frequency (Δf) of a vibrating structure that is subjected to small perturbations in the structural parameters i.e., effective stiffness or effective mass. Additional advantages of this method of detection are simple mechanical design, quasi-digital nature of the signal (thus using simple frequency measurement system such as frequency counter and not requiring additional analog-to-digital (A/D) conversion circuit), and ultra-high resolution [39,40,41], (up to 10–15 g scale [42,43,44] and up to 10–18 g scale [45,46,47,48]). Accurate frequency references, such as crystal oscillators, offer better stability than typical voltage or current references. Moreover, the output signal frequency of the resonant sensor is considered to be relatively immune to noise and interference [49]. However, they still inevitably suffer from environmental influences [50,51] and cannot satisfy the high-sensitivity requirement of future high-end markets such as ultra-small mass sensors [45] and ultra-low accelerometers [52]. It is fortunate that dual-resonator-based (DRB) and multiple-resonator-based (MRB) MEMS sensors have become an effective way to solve such issues due to their ability of sensitivity enhancement and rejection of environmental influences.

Nowadays, DRB and MRB MEMS sensors have attracted great attention from many researchers due to their advantages. Based on the relationship of resonators, DRB and MRB MEMS sensors can be categorized into three classes which include strength-coupled-resonator-based (SCRB) sensors [53,54,55,56,57,58,59,60,61,62,63,64,65,66,67,68,69,70,71,72,73,74,75,76,77,78,79,80,81,82,83,84,85,86,87,88,89,90,91,92,93,94,95,96,97,98,99,100,101,102,103,104,105,106,107,108], wave-coupled-resonator-based (WCRB) sensors [109,110,111] and uncoupled-resonator-based (URB) sensors [112,113,114,115,116,117,118,119,120,121,122]. 

The first kind of DRB and MRB MEMS sensors are the SCRB sensors, whose resonators are mutually coupled by coupling strength. In fact, the coupling strength can be exerted by mechanical structure, electrostatic force, and magnetic force. According to these coupling methods, SCRB sensors can be further classified into three types. The first type is a mechanical-strength-coupled-resonator-based (ME-SCRB) sensor. One of the advantages is that no further action is needed to couple those resonators together after the fabrication processes. The second type is an electrical-strength-coupled-resonator-based (EL-SCRB) sensor, of which the biggest merit is that the coupling strength can be easily tuned. The third type is a magnetic-strength-coupled-resonator-based (MA-SCRB) sensor, of which the virtue is that the coupling is not limited by the separation distance of those resonators. In recent ten years, those SCRB sensors have been comprehensively studied. Results have shown that strength-coupled resonators enable the SCRB sensors to function with merits of high sensitivity and high environmental stability. 

The second kind of DRB and MRB MEMS sensors are the WCRB sensors, of which resonators are acoustically coupled. Those resonators can be thought of as acoustic waveguides, where various acoustic waves, such as bulk acoustic waves, shear horizontal waves, surface acoustic waves and lamb waves can propagate. A variety of resonant modes can be generated by the propagation and reflection of guided waves in the resonator. Therefore, acoustic coupling can be formed by properly adjusting the structure of resonators. Moreover, higher sensitivity can be obtained by precisely measuring the beat frequency. In addition, some WCRB sensors also have the properties of low thermal expansion coefficient of frequency (TCF) due to those low TCF resonant modes and are not easily affected by external temperature interference. 

The last kind of DRB and MRB MEMS sensors are the so-called UCRB sensors, of which resonators are not coupled. Basically, they can be divided into three types based on the spatial relationship of resonators. The first type can be named as the DC-UCRB sensor because the resonators are directly connected. Higher sensitivity can be obtained by controlling the beat frequency to a small value. The second type is the dual-mode SRB sensor. It can be termed as quasi-UCRB (Q-UCRB) sensor due to the capability of simultaneously working on dual resonant modes. This mechanism of operation enables the Q-UCRB sensor to work with high sensitivity and low TCF properties. The last type can be called as PS-UCRB sensor, whose resonators are put together but physically separated. The frequencies of resonators will drift in the same or inverse direction in response to temperature changes when putting them in the same environment. Additionally, its drifts can be suppressed by the output of resonators with feasible algorithm or hardware. 

In a word, DRB and MRB MEMS sensors have the capabilities of sensitivity enhancement and environmental influences rejection. In view of the above advantages, it is worthwhile to summarize these DRB and MRB sensors to provide reference and guidance for future high-end market applications. With that purpose, we reviewed the-state-of-art of DRB and MRB MEMS sensors as follows: firstly, we analyzed single resonator and coupled resonator; secondly, a detailed introduction and analysis of DRB and MRB sensors, including SCRB sensors, WCRB sensors and UCRB sensors is given; finally, the future development of these sensors is prospected.

## 2. Analysis of Resonators

### 2.1. Single Resonator

The model of single resonator is shown in Figure 1a. The motion equation for such system can be denoted as:(1)$$M_{\mathrm{eff}}\ddot{x}+b_{\mathrm{eff}}\dot{x}+k_{\mathrm{eff}}x=F$$
where *M*_eff_, *b*_eff_, *k*_eff_, *F*, and *x* are the equivalent effective mass, effective damping constant, effective spring constant, the applied external force, and the motion displacement, respectively. Under ideal conditions, the frequency can be calculated by:(2)$$f=\frac{1}{2\pi }\sqrt{\frac{k_{\mathrm{eff}}}{M_{\mathrm{eff}}}}$$

Usually, a single resonator can be used to detect perturbation by the change in effective spring constant or/and mass. The relationship between frequency change and perturbation can be established on the basis of Equations (1) and (2). The normalized sensitivity can be obtained by [123]:(3)$$\left|S_{M}\right|=\left|\frac{\partial \omega_{0}}{\partial M}\times \frac{M}{\omega_{0}}\right|\approx \frac{1}{2}$$
(4)$$\left|S_{K}\right|=\left|\frac{\partial \omega_{0}}{\partial K}\times \frac{K}{\omega_{0}}\right|\approx \frac{1}{2}$$

However, a single resonator usually suffers from environmental influences, such as temperature, humidity, and pressure, which will result in unwanted shift of resonance frequency, calculated by [123]:(5)$$\Delta \omega =\frac{\Delta K+\Delta K_{\mathrm{environment}}}{2K}\times \omega_{0}$$

It is obvious that the unwanted frequency shift caused by environmental disturbance will greatly affect the performance of resonator. This disadvantage can be reduced by integrating dual or multiple resonators into a MEMS sensor, because all resonators have the same variation (common change) under the same environmental influences. In addition, the common change can be eliminated by appropriate algorithms or structure during the deduction of the sensor output.

The lumped element model is another method that can be used to analyze and simulate a single resonant system [124,125]. The damping (b), mass (M) and compliance (1/k) of each resonator in the mechanical domain can be modeled with motional resistance (*R*), inductance (*L*), and capacitance (*C*) in electrical domain (Figure 1b). On the basis of the electrical circuit, the motion equation can be transferred into:(6)$$L\frac{di_{m}}{dt}+Ri_{m}+\frac{1}{C}\int i_{m}dt=V$$
where *i*_m_ is the motional current, which presents the movement of the resonator, and *V* is the applied force of resonator. Therefore, the resonant frequency can be calculated by the classical *LC* resonant frequency Equation (7). The same result that a single resonator will be affected by the environmental influences can also be deduced in light of Equation (5).
(7)$$f=\frac{1}{2\pi }\sqrt{\frac{1}{LC}}$$

### 2.2. Coupled Resonators

#### 2.2.1. Dual Resonators Coupling

The mass-stiffness-damper model of two coupled resonators is shown in Figure 2. The coupling strength is denoted by the coupling ratio (*κ* = *K*_c_/*K*_eff_). It was discussed in [55] that the system is recognized as strong coupling when *κ* > 1 and weak coupling when *κ* < 0.1. However, when 0.1 < *κ* < 1, the weakly coupled and strongly coupled systems are not exactly defined. For weakly coupled resonators, frequencies of those two resonators are close, thus mode localization can form [126]. The output of the weakly coupled resonators is the eigenstates shift of those two resonators. However, due to the high spring constant of the coupling element, there is a difference for strongly coupled resonators, where the output of strongly coupled resonators is a frequency shift.

The frequency shift of strongly coupled resonators has been deduced in reference [125]. The mass-stiffness-damper model of coupled resonators can be described by the equivalent electrical circuit (Figure 3a), where the coupling part can be divided into mechanical force coupling, electrostatic coupling, and magnetic coupling according to the different coupling strength. It can be further simplified by substituting the coupler part with the T-shape capacitor network (Figure 3b). 

The whole circuit can be expressed with a 2-rank matrix:(8)$$\left[\begin{array}{cc} \left(\left(sL_{x}+\frac{1}{sC_{x}}+R_{x}\right)+\frac{1}{sC_{\mathrm{ca}}}+\frac{1}{sC_{\mathrm{cc}}}\right) & -\frac{1}{sC_{\mathrm{cc}}}\\ -\frac{1}{sC_{\mathrm{cc}}} & \left(\left(sL_{x}+\frac{1}{sC_{x}}+R_{x}\right)+\frac{1}{sC_{\mathrm{ca}}}+\frac{1}{sC_{\mathrm{cc}}}\right) \end{array}\right]\left[\begin{array}{c} i_{1}\\ i_{2} \end{array}\right]=\left[\begin{array}{c} V_{in}\\ V_{1} \end{array}\right]\;$$

The output of current over applied voltage can be obtained by solving this matrix.
(9)$$\frac{i_{1}}{V_{1}}=\frac{\sigma +\frac{1}{sC_{\mathrm{ca}}}+\frac{1}{sC_{\mathrm{cc}}}}{\left(\sigma +\frac{1}{sC_{\mathrm{ca}}}\right)\left[\sigma +\frac{1}{sC_{\mathrm{ca}}}+\frac{2}{sC_{\mathrm{cc}}}\right]},\;\mathrm{where}\;\sigma =sL_{x}+\frac{1}{sC_{x}}+R_{x}$$
(10)$$\frac{i_{2}}{V_{1}}=\frac{\frac{1}{sC_{\mathrm{cc}}}}{\left(\sigma +\frac{1}{sC_{\mathrm{ca}}}\right)\left[\sigma +\frac{1}{sC_{\mathrm{ca}}}+\frac{2}{sC_{\mathrm{cc}}}\right]}$$
where *C*_ca_, *C*_cb_, and *C*_cc_ are the equivalent capacitors of coupling part, respectively. According to Equations (9) and (10), the frequency shift (∆f) can be easily calculated.

The calculation of eigenstates changes in weakly coupled resonators have been clearly deduced in reference [123]. The normalized sensitivity of eigenstates changes can be expressed as:(11)$$\left|S_{M}\right|=\left|\frac{\partial \hat{u}}{\partial M}\times \frac{M}{\hat{u}}\right|\approx -\frac{K}{4\kappa }$$
(12)$$\left|S_{K}\right|=\left|\frac{\partial \hat{u}}{\partial K}\times \frac{K}{\hat{u}}\right|\approx -\frac{K}{4\kappa }$$
where, *u*, *M*, and *K* are the eigenvector, mass, and stiffness, respectively. We can easily observe that the sensitivity is improved by *K*/(2*κ*) times than that of single resonator on the basis of Equations (3) and (4) [123]. Results also revealed that the sensitivity of eigenstates changes will be two to three orders higher than that of frequency variations when the coupling strength is two or three orders smaller than the stiffness [56,74,123]. 

#### 2.2.2. Multiple Resonators Coupling

The mass-stiffness-damper model of three coupled resonators is shown in Figure 4. The output calculation of three coupled resonators is different from that of two coupled resonators, of which the frequencies can be calculated with [55]:(13)$$f_{1}=\frac{1}{2\pi }\sqrt{\frac{k_{\mathrm{eff}}}{M_{\mathrm{eff}}}};f_{2}=f_{1}\sqrt{1+\kappa };f_{3}=f_{1}\sqrt{1+3\kappa }$$
where *K*_eff_ = *K*_1_ = *K*_2_ = *K*_3_; *M*_eff_ = *M*_1_ = *M*_2_ = *M*_3_. And *f*_1_, *f*_2_, and *f*_3_ are the frequencies of three resonators, respectively. According to Equation (13), the frequency shifts of strongly coupled resonators array can be easily calculated.

For weakly coupled resonators array, the sensitivity of amplitude ratio can be denoted by [123]:(14)$$\left|S_{K}\right|=\left|\partial \right|\frac{X_{1}}{X_{3}}\left|/\partial \right|\frac{\Delta K_{\mathrm{eff}}}{K_{\mathrm{eff}}}∥=-\frac{K_{\mathrm{eff}}\left(K_{2}-K_{\mathrm{eff}}+K_{c}\right)}{K_{c}^{2}}\;\mathrm{When}\;K_{c}<\frac{K_{\mathrm{eff}\;}}{10}\;\mathrm{and}\;K_{2}>2K_{\mathrm{eff}\;}$$
where, *K*_2_, and *K*_c_ are the stiffness of resonator 2 and the coupling resonator, respectively. *X*_1_ and *X*_3_ are the movement of resonator 1 and 3, respectively. By comparing Equation (14) and Equation (12), we know that the normalized sensitivity of amplitude ratio of 3 weakly coupled resonators is improved by 4(*K*_2_ − *K*_eff_ + *K*_c_)/*K*_c_, which is at least 40 times higher than the normalized sensitivity of eigenstates changes of 2 weakly coupled resonators [123]. Other works [60,61] have also revealed that increasing the number of weakly coupled resonators is beneficial in improving its sensitivity. In order to be more intuitive and convincing, the comparison of single resonator, dual resonators and triple resonators are listed in Table 1.

## 3. DRB and MRB MEMS Sensors

DRB and MRB MEMS sensors can be realized by coupling resonators with strength, acoustic wave or by controlling resonators with feasible algorithm or hardware. On the basis of coupling methods, they can be categorized into three classes such as SCRB sensor, WCRB sensor and UCRB sensor. The following sections will analyze these three categories in detail. 

### 3.1. SCRB Sensors

The first kind of DRB and MRB MEMS sensors are the SCRB sensors, whose resonators are mutually coupled by coupling strength. In fact, the coupling strength can be exerted by mechanical structure, electrostatic force, and magnetic force. According to these coupling methods, SCRB sensors can be further classified into three types: mechanical-strength-coupled-resonator-based (ME-SCRB) sensors, electrical-strength-coupled-resonator-based (EL-SCRB) sensors, and magnetic-strength-coupled-resonator-based (MA-SCRB) sensors.

#### 3.1.1. ME-SCRB Sensors

The resonators of the ME-SCRB sensor are directly connected by a mechanical element, as can be seen in Figure 5a. The beam with dimensions of width W_c_, height H_c_, and length L_c_ (Figure 5b) can be denoted by the T-shaped capacitor network in electrical domain [127,128] (Figure 5c).

The impedance value of each capacitor can be calculated by [125]:(15)$$Z_{ca}=Z_{cb}=\frac{\cos\left(2\pi \cdot L_{c}/\lambda_{c}\right)-1}{jZ_{0}\sin\left(2\pi \cdot L_{c}/\lambda_{c}\right)}\;\mathrm{where}\;Z_{0}=\frac{1}{H_{c}W_{c}\sqrt{E\rho }}$$
(16)$$Z_{cc}=\frac{1}{jZ_{0}\sin\left(2\pi \cdot \lambda_{n}\right)}\;\mathrm{where}\;\lambda_{n}=\frac{v_{e}}{f_{0}}=\frac{\sqrt{E/\rho }}{f}$$
where *f*_0_ is the resonant frequency, *Z*_0_ and λ*_n_* are the normalized impedance and wavelength, respectively. *E* and *ρ* are the Young’s modulus and density of the resonator material, respectively. On the basis of Equations (15) and (16), the coupling capacitances of Equations (9) and (10) can be determined when the beam works at the extensional mode [129]. The resonant frequency is different from *f*_0_ through the addition or subtraction of C*_ca_* to or from the motional capacitor of the constitute resonator, *C_r_*. Then, the frequency shift with respect to the center resonant frequency *f*_0_ can be calculated by:(17)$$\frac{\Delta f}{f_{0}}=\sqrt{1+\frac{C_{r}}{C_{ca}}}-1≅\frac{C_{r}}{2C_{ca}}$$

It is obvious that the frequency shift between those two resonators can be controlled by changing the dimension of the coupling element.

Besides beam coupled resonators, other mechanical coupled resonators, such as corners coupled square resonators [130], middle side coupled Lame mode resonators, overhang coupling of beams and both-end connected same anchors [131], can also be used in ME-SCRB sensor [55,132] as shown in Figure 6. On the basis of these structures, resonators are either weakly or strongly coupled to form a ME-SCRB sensor.

Taking the coupling beam (Figure 6a) for example, we can define the coupling ratio (*κ* = *K*_c_/*K*_eff_) by the location relative to the vibration displacement. The coupling beam can be denoted by a matrix [55]:(18)$$\left[\begin{array}{ll} \frac{B\cdot c+C\cdot b}{B+b} & -\frac{2EI_{c}\alpha^{3}\left(B\cdot c\right)}{jwL_{C}^{3}\left(B+b\right)}\\ -\frac{jwL_{C}^{3}\left(C\cdot c-1\right)}{EI_{c}\alpha^{3}\left(B+b\right)} & \frac{B\cdot c+C\cdot b}{B+b} \end{array}\right]\left[\begin{array}{c} F_{2}\\ {\dot{x}}_{2} \end{array}\right]=\left[\begin{array}{c} F_{1}\\ {\dot{x}}_{1} \end{array}\right]$$
(19)$$\begin{array}{c} b=\sin\left(\alpha X\right),c=\cos\left(\alpha X\right),B=\sinh\left(\alpha X\right)\;\mathrm{and}\;C=\cosh\left(\alpha X\right)\\ X=\frac{x}{L_{c}}\;\mathrm{and}\;\alpha^{4}=\frac{\omega^{2}L_{c}^{4}\rho A_{c}}{EI_{c}} \end{array}$$
where *ω* is frequency, *x* is the vibration displacement, *A_c_*, *L_c_*, and *I_c_* are the cross section, length, and geometric moment of inertia of the coupling beam, respectively. *X* is the normalized distance from the center line. It can be deduced that the coupling ratio will become weaker when the coupling beam moves from the center toward the anchor rod, as shown in Figure 7.

During the last decade, a large number of ME-SCRB sensors have been developed such as mass sensors [55,56,57,58,68,69], electrometers [62,63] and acceleration measurement [15,16,64,65,66,70].

In 2006, Spletzer et al. [56] developed the first weakly coupled ME-SCRB sensor for mass detection (Figure 8). It can produce a significant change in up to 5–7% in the eigenstates while the relative change in the resonant frequency is only 0.01%, which illustrated that the sensitivity of eigenstate changes is 2 orders of magnitude higher than that of the frequency shift under ambient conditions. Ten years later, in 2016, Mohamad S. H. et al. [58] developed a strongly coupled ME-SCRB sensor for mass sensing (Figure 9). It is reported that the sensitivity of frequency shift of this sensor exceeds that of SRB sensor. Especially, the improvement in sensitivity is more than 20% when the coupling ratio equals 1.6. With the development of ME-SCRB sensors, in 2016, Honglong Chang et al. [62,63] developed a weakly coupled ME-SCRB sensor for electrical potential measurement (Figure 10a). Results showed that the sensitivity of eigenstate changes was improved by three orders of magnitude higher than that of frequency changes. Two years later, in 2018, Honglong Chang et al. [64] successfully developed a weakly coupled ME-SCRB sensor to measure acceleration for the first time (Figure 10b). The sensor exhibited good properties by using amplitude ratio shift as output. Results showed that the sensitivity of amplitude ratio (~312,162 ppm/g) is approximately 302 times higher than that of the frequency change (~1035 ppm/g).

It can be concluded from the above-mentioned research works that ME-SCRB sensors with dual resonators have exhibited good performance. Besides, ME-SCRB sensors with three resonators have also been developed. In 2018, Honglong Chang et al. [66] successfully developed a weakly coupled ME-SCRB sensor with three resonators to measure acceleration for the first time (Figure 10c). In this sensor, resonators are weakly coupled by two coupling beams. When inertial perturbation acts on the proof mass, the sensor will produce shifts in amplitude ratio due to the mode localization between the two outer resonators. It is reported that the sensitivity of amplitude ratio of this accelerometer is improved by 348% in comparison with that of eigenstate changes of the accelerometer. Other ME-SCRB sensors with no less than three resonators have also developed. In 2008, Spletzer et al. [57] fabricated a fifteen-cantilever array as a weakly coupled ME-SCRB sensor (Figure 11a). It is reported that the magnitude of eigenvector has a large relative change in the order of 10–100% due to the added mass, while the highest relative frequency shift is only about 0.1%. The relative shift of eigenvector was two to three orders higher than the relative frequency shift. Six years later, in 2014, Mohamad S. H. et al. [55] reported a cantilever-array-based strongly coupled ME-SCRB sensor for mass measurement (Figure 11b). In the second mode of vibration, a significant frequency shift can be observed in this sensor, which also proved that the sensitivity of cantilever-array-based strongly coupled ME-SCRB sensor exceeds that of a SRB sensor.

The comparisons of the properties of different ME-SCRB sensors are listed in Table 2. We can draw the conclusion that the strong or weak coupling has enabled the ME-SCRB sensors a good performance. According to the abovementioned works, we can also conclude that one of the biggest advantages of ME-SCRB sensors is that, once resonators are connected, no further action is needed to couple them together. Although the tuning of the coupling strength is difficult, the ME-SCRB sensors are promising in the field of high-end sensors. However, deviations may occur in the device due to the fabrication errors. Hence, researchers are still sparing no effort to optimize these ME-SCRB sensors

#### 3.1.2. EL-SCRB Sensors

Resonators of EL-SCRB sensors are mutually coupled by electrostatic force. Similarly to the coupling element of mechanically coupled resonators, the coupling capacitor of electrically coupled resonators can also be substituted with the T-shape capacitor network in the lumped element model. Hence, the frequency change can be calculated following the Equations (8)–(10). However, the frequency shift is not good enough to describe the properties of weakly coupled resonators. In 2016, Chun Zhao [123] summarized the deduction processes of the shift of eigenstates for two and three coupled resonators and that of the shift of amplitude ratio for three coupled resonators. Basically, all of the the EL-SCRB sensors are developed on the basis of these theoretical results. Electrically coupled resonators were first introduced in MEMS filter by Pourkamali and Ayazi [76] in 2005. Four years later, in 2009, the first EL-SCRB sensor was designed by the group of A. Seshia [74].

From 2010 to 2012, the group of A. Seshia have developed several EL-SCRB sensors including mass sensor [77], displacement sensor [78] and electrometer [79] as shown in Figure 12a–c. Results illustrated that relative shift of eigenstates of EL-SCRB sensor for mass sensing [77] is about 4.32% and 3.448% at the two eigenvalues, respectively, while the relative variation of frequency is only about 0.00237%, which proved the much higher sensitivity of eigenstates shift than that of the frequency shift. Such conclusion has also been proved in EL-SCRB sensors for displacement measurement [78] and charge sensing [79] application. The relative shifts of eigenstates are two or three orders of magnitude higher than that of frequency shift. Recently, M. Lyu [80] proposed a novelty mass sensor with distributed electrodes based on the mode localization of two electrostatically coupled microbeams in higher-order modes. Compared to the output metric of frequency shift, the sensitivity improved by four orders of magnitude by using the amplitude ratio as the output.

The comparisons of properties of different EL-SCRB sensors are listed in Table 3. In addition to the high sensitivity property, these EL-SCRB sensors can also obtain the capability of common mode rejection, because the eigenstates of those coupled resonators are affected to the same extent.

The EL-SCRB sensors with three resonators are also studied. From 2015 to 2018, Chun Zhao et al. have developed several EL-SCRB sensors with three resonators (Figure 13) for mass [81], stiffness [84,85,86] and force [87] sensing applications. Results showed that when the EL-SCRB sensor is used for mass sensing application, the sensitivity of the change in the amplitude ratio is around two orders of magnitude larger than that of frequency shift and is about twice of amplitude change. When the EL-SCRB sensor is used as a stiffness sensor [84], the highest normalized sensitivity is 13,558, which has been improved at least 56 times than that of the EL-SCRB sensor with 2 resonators. The further application of EL-SCRB sensor in force detection [87] has also been studied. Results showed that the sensitivity is two orders of magnitude higher than that of the conventional single resonator force sensor. The comparisons of the EL-SCRB sensors for mass, stiffness, and force detections are listed in Table 4. We can draw the conclusion from the data of Table 4 that the sensitivities of the EL-SCRB sensors have been much improved compared with those of single resonator sensors, and increasing the number of coupled resonators is beneficial to further improve its sensitivity, which is the great merit of multiple coupled resonators.

Through a comprehensive analysis of these studies of EL-SCRB sensors, we can observe that the biggest advantage of this kind of sensor is that the coupling strength can be easily tuned. However, when the difference between the resonant frequencies of the two resonators is too small, the two resonant modes will interfere with each other. This phenomenon, which is termed as mode-aliasing [87], should be avoided in the sensor design process. The anti-mode-aliasing condition has been found in reference [87]. Reference [87] also pointed out that the impractical for real time application, low Q-factor resonators and high noise are the three limitations of EL-SCRB sensors. Fortunately, Chun Zhao et al. have been working on resolving those above problems. In 2018, they proposed a noise optimization method [88] with an optimal region for the resolution existing in vicinity of amplitude ratio 1.22. However, a trade-off between resolution and linearity still exists as the optimal region is not the linearity part.

#### 3.1.3. MA-SCRB Sensors

Resonators of MA-SCRB sensors are mutually coupled by magnetic force. Generally, the distance between electrically and mechanically coupled resonators is in the range of 1 nm to 1 μm. From the fabrication point of view, it is hard to uniformly manufacture them due to the uneven manufacturing processes. In order to resolve such problem, Pai et al. [91] developed a MA-SCRB sensor, which is used as gyroscope with large separations. In this sensor, neodymium permanent magnets, installed on the proof mass, are used to form a magnetic coupling, as shown in Figure 14b. The actuation and sensing of the resonators are realized through electrostatic transduction, as shown in Figure 14c. In summary, the MA-SCRB sensor is feasible. However, much work needs to be conducted before it can be put into practical use.

#### 3.1.4. Common Mode Rejection of SCRB Sensors

The ability of common change rejection of coupled resonators has been investigated. Reports showed that environmental factors (such as ambient pressure [90,91] and/or temperature [92,93,94,95]) or nonspecific bindings (for mass sensing) influence all of the vibrating elements uniformly, ideally not affecting the eigenmodes of the system, while shifts in the resonance frequencies still occur. One of the early works addressing the common-mode rejection of mode-localized sensor is provided in [90]. Experimental results demonstrated its intrinsic common-mode-rejection ability. Other work in [91] also demonstrated the ambient pressure drift rejection capability of the mode-localized sensors. Results showed that amplitude ratio (AR) based output remained relatively insensitive against the ambident pressure drift compared with frequency shift output. The maximum error of AR based output in sensitivity was 2.74%, whereas that of the frequency shift output reached to about 21.6% for a pressure range of 2.6 to 20 Pa. In addition to ambient pressure, mode-localized sensors also have the ability to suppress the impact of environmental temperature on the sensors. Results in [92] presented that when temperature changes, maximum measurement error in the AR output is 8.8% whereas maximum error in measurement of frequency shift output is >1000%. Results in [93] also demonstrated the ability of mode-localized sensor in immunity to temperature fluctuations (between 35 °C and 60 °C). Other works [94,95] also revealed that the AR has an excellent temperature drift suppression capability compared to the frequency shift output. The reason for the proposed advantage of using AR shift output is that any ambient variable (for instance temperature/pressure) will equally affect the output sine waveform of each of the resonators. Therefore, environmental effect is cancelled to the first order with the ratio-based output.

#### 3.1.5. Resolution of SCRB Sensors

Owing to the mode localization phenomenon of weakly coupled resonators (WCRs), the sensitivity of the amplitude ration (AR) output metric is at least two or three orders of magnitude higher than that of the resonant frequency, which has been theoretically and experimentally demonstrated by various types of WCRs-based resonant sensors. Nevertheless, to what extent the mode-localized sensing paradigm influences the ultimate resolution remains unknown. Additionally, this idea even was regarded as the most important fundamental research question in this field in 2018. 

Zhang et al. reported a high-sensitivity resonant electrometer [63] based on the mode localization of two degree-of-freedom (DoF) WCRs. The experimental results showed that the sensitivity enhancement is more than three orders of magnitude when selecting the amplitude ratio instead of the resonant frequency as the output metric. However, the resolution of the electrometer using both frequency and amplitude ratio readouts is in the same range. They also reported an acceleration sensing method [65] based on two WCRs and demonstrated that the measured relative shift in amplitude ratio was 302 times higher than the shift in resonance frequency but the improvement in the resolution was not obvious. M. Pandit [73] explored coupled nonlinear MEMS resonators as sensors based on the principle of energy localization, an improvement of 2× in input-referred noise floor, or resolution, in comparison to its linear counterpart at both the bifurcation points, has been achieved around veering and an improvement of 4× at the top bifurcation point away from veering despite of reduced sensitivity. This is likely due to the higher Signal to Noise Ratio (SNR) and additional noise filtering properties at the bifurcation points. 

Regarding the problem of resolution of mode-localized sensors, there are mainly two different views. J. Juillard et al. [96,97] hold the view that mode-localized sensors based on the amplitude ratio output metric provide measurements whose resolution is independent on coupling strength, while A. Seshia [98] and Chang [99] believe that that the resolution of mode-localized sensors is dependent on the strength of internal coupling *k*. They believe that weaker effective coupling (lower *k*) between the resonators should help enhance not only the sensitivity of such sensors, but also contribute to substantial improvements in the resolution. Different views of resolution limit models are shown in Table 5. 

Chang [99] also pointed out that the resolution of AR-based mode-localized sensors can be improved by increasing DoFs in the resonant system, and the resolution limit of the high-order (the DoFs > 3) mode-localized sensors using the AR output metric is better than that of the frequency-output metric, as shown in Table 6. They revealed that the 3-DoF mode-localized sensors using the AR output metric have a better resolution limit than both the 2-DoF and frequency-output sensors, and the 4-DoF sensors indicate the best resolution limit, which is three orders better than 3-DoF. This trend coincides with the experimental data in [98,104,105], of which the resolution is improved from 8000 e/√Hz (2DoF) [100] to 9.21 e/√Hz (3-DoF) [100] and 0.256 e/√Hz (4-DoF) [101].

Overall, the resolution of mode-localized sensors is related to multiple factors such as sensitivity, noise, and quality factors; we can improve the resolution by further increasing the sensitivity and reducing the amplitude noises both of extrinsic noise arising from the external electronic interfacial readout circuitry, and intrinsic noises that are inherent to the micro- or nano-mechanical resonator arrays.

#### 3.1.6. Summary

In this section, three different types SCRB sensors, named as ME-SCRB sensor, EL-SCRB sensor and MA-SCRB sensor, were introduced in detail, respectively. In general, all of these SCRB sensors have the merit of high sensitivity and common change rejection, but the resolution improvement is not obvious, and there is still a lot of room for enhancement. Besides, there are still other more or less shortcomings that need to be resolved. The ME-SCRB sensor has the disadvantage of bad tunability of coupling strength; the EL-SCRB sensor has the weak points of low Q-factor resonators and high noise; the MA-SCRB sensor is too far away from real application. In addition, they are all limited by the fabrication processes. A good fabrication process can accurately control the coupling strength of the ME-SCRB sensor, improve the Q-factor of the EL-SCRB sensor, and reduce the non-uniformities of magnets fabrication in the MA-SCRB sensor. Therefore, it is necessary to spare no effort in the precision manufacturing process to make these sensors get practical applications in the future.

### 3.2. WCRB Sensors

The second kind of DRB and MRB MEMS sensors are the WCRB sensors, of which resonators are acoustically coupled. Generally, the resonators of WCRB sensor can be thought as acoustic waveguides [109,110,111]. The dimension-related properties of waveguides determine the types of propagated acoustic waves. It offers an opportunity for different resonant modes which can be coupled together with the same frequency. In 2013, Roozbeh Tabrizian et al. [109] proved that due to the dispersion characteristics of the Lamb wave, different resonant modes of a Lamb wave resonator (LWR) can be coupled into a whole sensor structure by engineering the width of resonator (Figure 15) [109]. Results in [109] showed that by extracting a small beat frequency from an integer combination of two resonance modes with large difference in their temperature coefficient of frequency (TCF), a high TCF of ~8300 ppm/°C was achieved, which demonstrated the suitability of acoustically coupled resonators for temperature sensing with high accuracy and resolution.

For other specific LWR that vibrates in the coupling of first-order width extensional mode (WE1) and the second-order width shear mode (WS2), the width of each part, (*W*_1_, *W*_2_) can be calculated based on the same frequency of different parts: (20)$$\begin{array}{l} f_{1}=f_{2}\Rightarrow \frac{V_{2}}{W_{2}/2}=\frac{V_{1}}{W_{1}}\Rightarrow \frac{W_{2}}{W_{1}}=\frac{2V_{2}}{V_{1}}\;\mathrm{where}\;V=\sqrt{\frac{C}{\rho }}\\ \Rightarrow \frac{W_{2}}{W_{1}}=\sqrt{\frac{4C^{\prime }}{C^{\prime \prime }}}=\sqrt{\frac{4\left(C_{11}-C_{12}\right)}{C_{11}+C_{12}+2C_{44}}} \end{array}$$
where *C*′ is the quasi-shear elastic constant in shear modes and *C*″ is the quasi-longitudinal elastic constant in extensional modes.

Low temperature coefficient of frequency (TCF) is one of the merits of the WCRB sensor, which can be realized by coupling low temperature coefficient modes with other modes that are sensitive to physical/environmental signals. It has been proved that the shear mode has a lower TCF than the other mode. Therefore, a low TCF resonant sensor can be developed by coupling shear mode with other modes. In 2013, Roozbeh Tabrizian et al. [110] developed a resonator, of which the extensional and shear mode are acoustically coupled through evanescent wave in the intermediate region. The central part (*W*_1_) of the sensor works in WE1 mode while the edge part (*W*_2_) resonates at the WS2 mode with the same frequency of central part. Results showed that compared with SiBAR, the WCRB sensor has achieved considerable Q enhancement and TCF reduction.

Besides the low TCF characteristics, high sensitivity is another merit of WCRB sensor, which can be obtained by precisely deducing the beat frequency. After developing the shear-extensional mode coupled resonator, Roozbeh Tabrizian et al. developed a WCRB sensor for pressure measurement with high sensitivity in 2014 [111], as shown in Figure 16. In this sensor, 2 silicon bulk acoustic resonators (SiBAR) are excited in their 3rd length-extensional mode (LE_3_) and are acoustically coupled through thin vertical membranes. When out-of-phase bulk acoustic wave propagates into vertical membranes, the vertical membranes is excited to work at extensional Lamb mode as shown in Figure 17. Moreover, when in-phase bulk acoustic wave propagates into the vertical membranes, the vertical membranes is excited to work at transverse flexural resonant mode. The transverse flexural mode is sensitive to their surrounding air molecule while the extensional mode is not. Hence, a pressure sensor with an amplified sensitivity can be developed by coupling the two resonant modes. In such WCRB sensor, the beat frequency *f*_b_ is used to describe the pressure changes and can be expressed with:(21)$$f_{b}\approx \frac{1}{2\pi }\left|\sqrt{\frac{K_{{\mathrm{LE}}_{3}}+\alpha^{2}K_{fk}}{M_{{\mathrm{LE}}_{3}}+\alpha^{2}\left(M_{flx}+\Delta M\left(P\right)\right)}}-\sqrt{\frac{K_{{\mathrm{LE}}_{3}}}{M_{{\mathrm{LE}}_{3}}}}\right|$$
where ∆*M*(*P*) is the Pressure sensitive mass, α is the acoustic coupling efficiency, M_i_ and K_i_ (i ϵ {LE_3_, *flx*}) are the equivalent mass, spring of LE_3_ mode in SiBARs, and extensional and flexural modes in vertical membranes, respectively. In fact, the length of thin vertical membranes (2L) is 2 times than the length of SiBARs, which makes the LE_3_ mode of SiBARs and extensional Lamb mode of vertical membranes to resonate at the same frequency but a different wavelength. Results showed that a high sensitivity of 346 ppm/kPa has been achieved in the pressure range of 0–100 kPa, which is more than 500 times higher than that of pressure sensor in [6], whose sensitivity is 0.69 ppm/kPa.

Comparisons of WCRB sensors are listed in Table 7. It can be observed that high sensitivity and low TCF can be obtained through proper acoustic coupling. Although WCRB sensors are currently only proof-of-concept, they have been proven to be promising for sensing applications and can be further enhanced by optimizing measurement and design methods.

### 3.3. UCRB Sensors

UCRB sensors are the last type of DRB and MRB MEMS sensors. In fact, some UCRB sensors have two or more uncoupled resonators, while others have only one resonator. Those UCRB sensors can obtain the ability of high sensitivity and low TCF by properly controlling the resonators. On the basis of the spatial relationship of resonators, UCRB sensors can be mainly classified into three types: direct connection (DC-UCRB) sensors, quasi dual resonator (Q-UCRB) sensors and physically separated (PS-UCRB) sensors.

#### 3.3.1. DC-UCRB Sensors

The first type of UCRB sensors is the so-called DC-UCRB sensors. Although connected by physical structures, those resonators are not coupled to each other. One of the resonators is used as a sensor, while the other resonator is used as a reference. Study [112] has shown that based on the beat frequency of the two resonators, the DC-UCRB sensor can achieve excellent temperature sensing performance. Besides high sensitivity, the dual-mode oscillator can realize self-temperature compensation and avoid the temperature hysteresis effect caused by the external temperature compensation sensor, which permits the realization of high-accuracy Microcomputer-Compensated Crystal Oscillators (MCXO).

Usually, a beat frequency *f*_B_ is used as the output and can be defined as:(22)$$f_{B}=m\times f_{1}-f_{2}$$
where *m* is the frequency multiplication factor, *f*_1_ and *f*_2_ are the first and the second resonant frequency, respectively. Normally, *f*_1_ should be less than *f*_2_. It has been proved that high sensitivity performance can be realized by designing the beat frequency to be as small as possible compared with the natural frequency of the resonator. Thus, it is best to make the value of m as close as possible to the frequency ratio r = *f*_2_/*f*_1_. 

In 2017, H. Campanella et al. [113] developed a DC-UCRB sensor for temperature sensing as shown in Figure 18b. It consists of two resonators, of which the frequencies are widely separated. The frequency of the outer resonator is 180 MHz while that of the inner resonator is 500 MHz. Each resonator is excited in fundamental symmetric Lamb-wave modes (S0). Results showed that when the beat frequency is designed as 11.5 MHz, the TCF of the beat frequency (−334 ppm/°C) is much larger than the separately extracted first-order TCFs of the outer (−30 ppm/°C) and inner resonator (−23 ppm/°C), which proved the UCRC sensor’s suitability a for highly sensitive temperature sensor.

Generally, DC-UCRB sensors are promising in designing high-sensitivity sensors. However, due to the transverse wave propagation between two resonators, it has not become a hotspot in the research field.

#### 3.3.2. Q-UCRB Sensors

The Q-UCRB sensors are the second kind of UCRB sensors. In fact, Q-UCRB sensors are essentially SRB sensors, with only one resonator working in dual or multiple modes [114,115,116,117] by properly controlling the excitation circuit. Although there is only one resonator, the single resonator working in dual modes can play the same role as dual resonators. Hence, the merits of dual resonators sensors can be inherited in Q-UCRB sensors. To ensure that the Q-UCRB sensor can work at dual modes, we should carefully design the excitation circuit on the basis of Barkhausen’s criteria [119]. In addition, to achieve high sensitivity performance, it is necessary to carefully design the beat frequency calculation algorithm.

In 2011, the group of Farrokh Ayazi fabricated a Q-UCRB sensor for high-performance temperature sensing, of which the resonator vibrates at its fundamental (*f*_1_) and third-order (*f*_3_) length extensional modes as shown in Figure 19a [114]. The frequencies of *f*_1_ and *f*_3_ are 30 MHz and 87 MHz, as shown in the top of Figure 20, respectively. When it is simultaneously driven by two loops (bottom left of Figure 20), the resonator will produce an output, which is a superposition of two signals. Results showed that a beat frequency (*f*_b_) with a TCF of 162 ppm/°C was obtained. Three years later, they [115] further enhanced the performance of Q-UCRB sensor by simultaneously exciting the in-plane width-shear (WS) and width-extensional (WE) modes of the silicon resonator. Results showed that when the beat frequency is equal to 17 kHz, the highest TCF of the beat frequency is 1480 ppm/°C, which is almost 10 times that of the Q-UCRB sensor developed in 2011.

Besides the abovementioned works, they [109] developed another Q-UCRB sensor for temperature sensing in 2013. As shown in Figure 21, the central part of the resonator is designed to operate in the in-plane lamb wave mode. On the basis of the dispersion characteristic, the out-of-plane mode can also be excited in the central part by adjusting the width of the interdigital electrodes. Thus, the Q-UCRB sensor can run at multiple resonant modes simultaneously and independently. Particularly, the integer frequencies ratio of in-plane mode and out-of-plane mode can generate a small beat frequency and thus an amplified temperature sensitivity. Results showed that the TCF of beat frequency is 8292 ppm/°C, which is a big advancement in comparison to the abovementioned work of this group.

Other groups have also carried out research on the Q-UCRB sensor. In 2012, L. Garcia-Gancedo et al. [116] developed a dual-mode thin film bulk acoustic wave resonator (FBAR) which can precisely measure the mass loading and the temperature in parallel, without the need for additional reference devices or complicated electronics for real-time temperature compensation. In 2018, Congcong Gu et al. [117] proposed a dual-mode film bulk acoustic resonator (FBAR) based pressure sensor. The feasibility of on-chip temperature compensation of such sensor has been proved by the experimental results. However, it still cannot be used in practical applications because it is difficult to design a circuit to enable the sensor to simultaneously work in two resonant modes.

The comparisons of different Q-UCRB sensors are listed in Table 8. We can draw the conclusion that Q-UCRB sensors usually have high sensitivity and can realize temperature compensation without external reference. More importantly, they can detect multiple parameters simultaneously.

#### 3.3.3. PS-UCRB Sensors

The last type of UCRB sensors is PS-UCRB sensors [118,119,120]. Resonators of PS-UCRB sensors are separately located in the same environment and will drift in the same or inverse directions in response to changes through appropriate design. The temperature-induced resonant frequency shift needs to be suppressed by feasible algorithm or hardware.

In 2015, the group of Junbo Wang [118] developed a PS-UCRB sensor for pressure measurement with the ability of self-temperature compensation. As shown in Figure 22, there are two “H”-shaped double-clamped beams, which are physically separated by gaps, named as central beam and side beam. With almost identical dimensions and comparable resonant frequencies, the resonant frequencies of those two beams drifted in the same direction in response to temperature changes. Experimental results showed that the pressure sensor has good properties with error less than ±0.01% of the full pressure scale (50 kPa~110 kPa) in the entire temperature range (−40 °C~70 °C).

In 2017, Qingyun Xie et al. [119] developed a PS-UCRB sensor for high pressure measurement in high temperature environment as shown in Figure 23. Two methods are adopted to reduce the temperature influence: the first is to make an oxide trench array (OTA) in the poly-Si layer, and the second is to design algorithm based on dual-frequency resonating device. It was reported that a high-pressure sensor with a non-linearity of 2.28% F.S. has been obtained over a wide temperature range of 360 °C.

In the abovementioned PS-UCRB sensor, the temperature-induced resonant frequency shift is suppressed by algorithm. In addition to temperature compensation algorithms, temperature compensation can also be implemented through hardware. SiTime Company [120] developed a high-resolution PS-UCRB sensor for temperature measurement (Figure 24a), of which the temperature information is deduced by hardware. It is shown that two pairs of four-ring-resonator structure are designed with inverse response and different sensitivity of temperature. A hardware-based approach of measuring the frequency ratio of two on-chip four-ring-resonator structure (Figure 24b) is designed [121,122] to generate high temperature resolution. The output frequency can be expressed by:(23)$$f_{\mathrm{out}}=\left[\left(1+TDC\right)\times PFM\right]\times f_{\mathrm{ref}}/N_{\mathrm{div}}$$
where *f*_out_ and *f*_ref_ are the frequencies of output and PLL reference clock frequency, respectively. *TDC* is the frequency ratio of two on-chip resonators, *PFM* is the programmable frequency multiplier, and *N*_div_ is the value of divider. 

The block diagram of temperature measurement with the PS-UCRB sensor is shown in Figure 25. The first resonator operates at 47 MHz which exhibits a less than ±50 ppm frequency stability in the temperature range of −45 °C to 105 °C. The second resonator oscillates at 45 MHz with a temperature coefficient of −7 ppm/K. Results showed that the PS-UCRB sensor have a very high resolution of 20 μK over a bandwidth of 100 Hz. Now, this work has been successfully applied to the suppression of temperature-caused resonant frequency shift in commercialized timer chips.

The comparisons of different PS-UCRB sensors are listed in Table 9. We can see that high resolution and low TCF can be obtained. The virtues of PS-UCRB sensors are as follows: (1) using two separated resonators for good temperature compensation and (2) high resolution of temperature sensing.

#### 3.3.4. Summary

In this section, three different types UCRB sensors, named as the DC-UCRB sensor, the Q-UCRB sensor and the PS-UCRB sensor, were introduced in detail, respectively. In general, these UCRB sensors have the merits of high sensitivity, parallelize measurement, and common change rejection. Yet, there are still shortcomings that need to be resolved, such as how to simultaneously excite multiple oscillator circuits and how to optimize the signal-to-noise ratio and the linearity of the beat frequency.

## 4. Conclusions

Dual-Resonator-Based (DRB) and Multiple-Resonator-Based MEMS Sensors are comprehensively and critically reviewed in this paper. The review started with understanding the principle of single resonator and followed by the analysis of coupled resonators based on the principle of mode-localization. Merits of sensitivity enhancement of coupled resonator using vibration amplitude ration (AR) as output compared with frequency shift output are highlighted. Following the analysis of coupled resonators, three different classes DRB and MRB MEMS sensors, named strength-coupled-resonator-based (SCRB) sensors, wave-coupled-resonator-based (WCRB) sensors, and uncoupled-resonator-based (URB) sensors are introduced and analyzed in detail, respectively. Several sensing applications, such as mass sensor, displacement sensor, electrometer, accelerometer, stiffness sensor, and force sensor are given. A comparative performance between DRB and MRB sensors and single-resonator-based sensors are also provided in tables to demonstrate their advantages of sensitivity enhancement. Studies revealed that weaker effective coupling (lower *k*) between the resonators can help further enhance the sensitivity of such sensors. Besides high sensitivity, the ability of common mode rejection is another advantage of DRB and MRB MEMS sensors, which are also analyzed throughout the study. Results showed that AR has an excellent disturbance suppression capability compared to the frequency shift output. Following the analysis of common mode rejection, another sensor’s performance characterization, resolution, are discussed and analyzed. Regarding whether the resolution of the mode local sensor is related to the coupling coefficient *k*, there are mainly two different views. Anyhow, the resolution of mode-localized sensors is related to multiple factors such as sensitivity, noise, and quality factors, we can improve the resolution by further increasing the sensitivity and reducing the amplitude noises both of extrinsic noise arising from the external electronic interfacial readout circuitry, and intrinsic noises that are inherent to the micro- or nanomechanical resonator arrays.

Overall, DRB and MRB MEMS sensors have many advantages over SRB sensors, such as high sensitivity and common mode rejection. These advantages make them attractive for many applications in high-end markets, such as the measurement of ultra-small mass sensors and ultra-low accelerometers. This review assessment is proposed to serve as a one-stop reference for other researchers in the field to access a comprehensive outlook on the promising DRB and MRB MEMS sensors.

## 5. Future Perspectives of DRB and MRB Sensors

Owing to their inherent merits of high sensitivity and common mode rejection, DRB and MRB MEMS sensors are promising in future high-end markets. However, a lot of work still needs to be conducted to bridge the gap between sensor research and market applications, especially in the following two aspects.

Firstly, fabrication processes are the key factor in the realization of DRB and MRB MEMS sensors, which need to be further optimized. For SCRB sensor, the realization of high stability and high Q factor resonators depends on the degree of sealing of the device in a vacuum environment. The better the vacuum environment, the higher of the quality factor. Obviously, based on the Leeson model [135], a high Q is very important to reduce the phase noise level of the resonator. For the WCRB sensor, precise fabrication of the structure is the key factor to complete the coupling of different dimension-related resonant mode. Moreover, it can also further reduce the leakage of acoustic wave from the edge of the resonant structure to greatly improve the precision of resonant frequency, and ultimately enhance the stability of frequency. For the DC-UCRB and the Q-UCRB sensor, high fabrication precision determines the sensitivity and the linearity of beat frequency due to the piezoelectric transduction principle. For the PS-UCRB sensor, the fabrication of same resonant structure is the most important factor, which is key in obtaining high sensitivity. Therefore, the manufacturing process needs to be further improved in the future.

Secondly, the control circuit needs to be further studied. The groups of Honglong Chang [73] and A. A. Seshia [136] have contributed a lot for the self-oscillating circuit loop of SCRB sensor. On the basis of these developed self-oscillating closed loops, the SCRB sensors have achieved the abilities of high sensitivity, high resolution, and common mode rejection. Especially, the high signal-to-ratio noise (124.2 dB) of self-oscillating closed loop has been obtained by group of Honglong Chang. Such work has greatly improved the development of DRB and MRB MEMS sensors. Besides, the frequency stability needs to be further enhanced. So far, the best frequency stability achieved by A. A. Seshia group is 7.5 ppb. There are some effective ways to improve frequency stability [136], such as the optimal operating point of the oscillator, low power consumption and integrated implementation of the oscillator, as well as integration with a temperature compensation scheme. For WCRB and UCRB sensors, besides the PS-UCRB sensor, the optimization of signal-to-ratio noise and the linearity of beat frequency is the imperative work. For the PS-UCRB, the optimization of circuit should be conducted by combining the design of algorithm.

All in all, dual-resonator-based (DRB) and multiple-resonator-based (MRB) sensors are indeed promising in future high-end markets, but there is still a long way to go before realizing their practical applications in the field of sensing.

## Figures and Tables

**Figure 1 micromachines-12-01361-f001:**
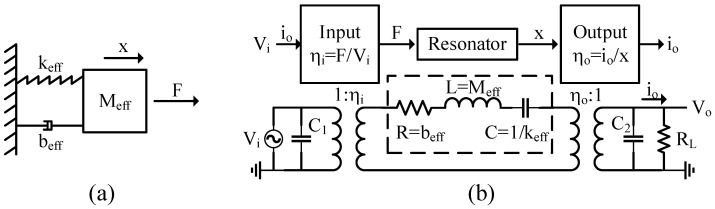
(**a**) A simple resonator model; (**b**) the block diagram and the electrical circuit of a resonator.

**Figure 2 micromachines-12-01361-f002:**
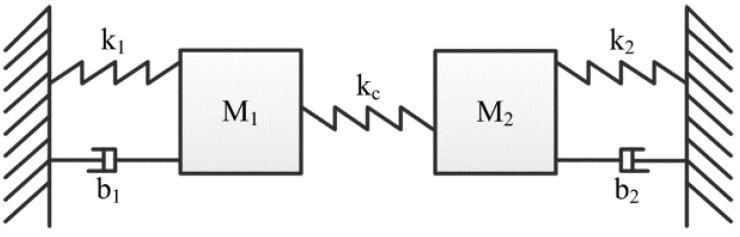
The mass-stiffness-damper model of coupled resonators.

**Figure 3 micromachines-12-01361-f003:**
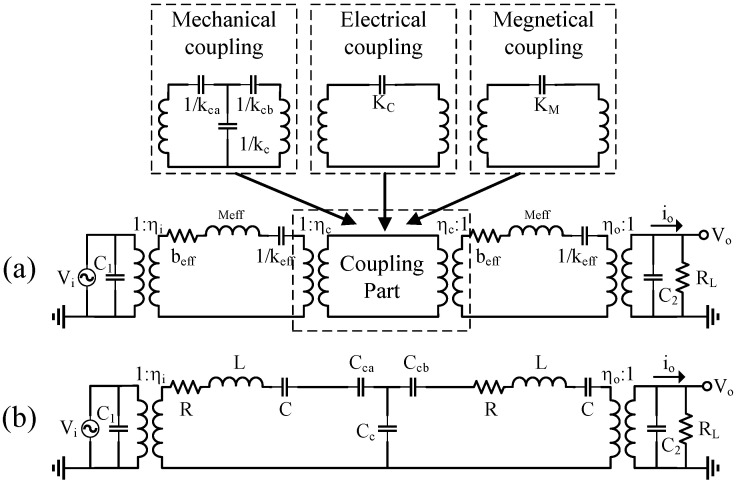
(**a**) Equivalent electrical circuit; (**b**) a simplified version of the circuit.

**Figure 4 micromachines-12-01361-f004:**
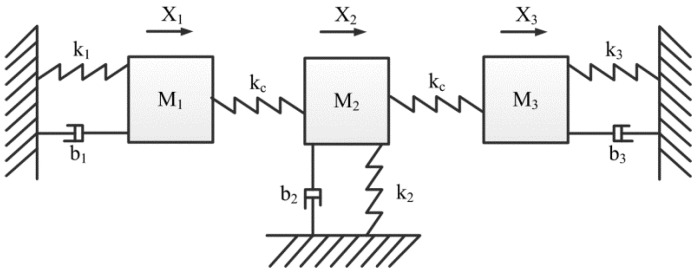
The mass-stiffness-damper model of three coupled resonators.

**Figure 5 micromachines-12-01361-f005:**
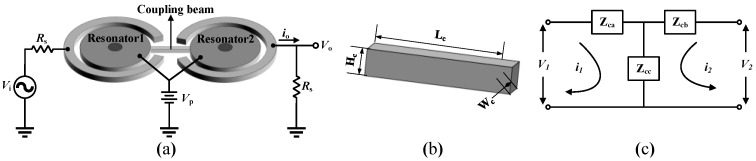
(**a**) Figure of mechanically coupled resonators; (**b**) Dimension of the coupling beam; and (**c**) T-shaped capacitor model for the coupling beam.

**Figure 6 micromachines-12-01361-f006:**
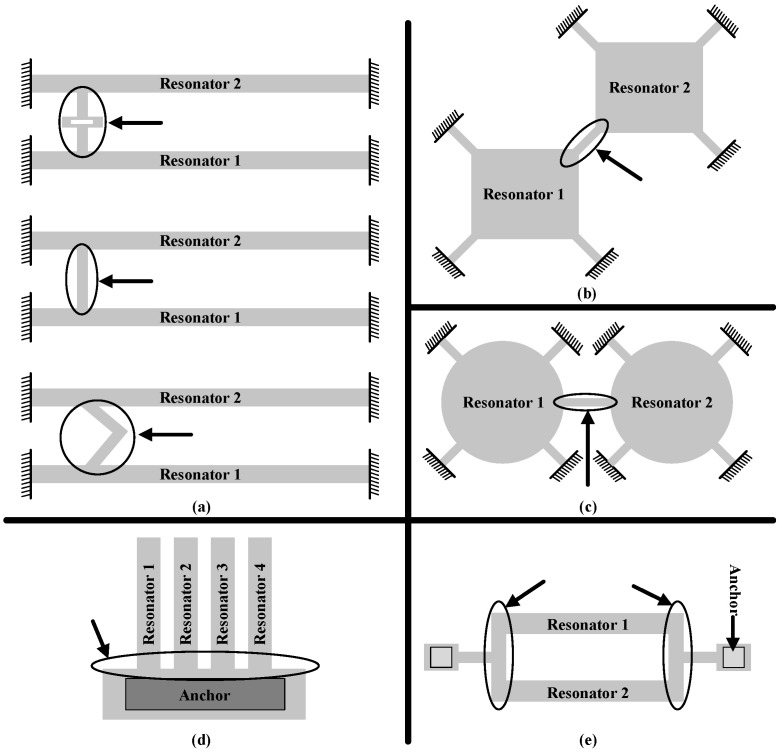
Structures of mechanical coupling of resonators. (**a**) Beam coupled; (**b**) Corner coupled; (**c**) Middle side coupled; (**d**) Overhang coupled; and (**e**) Double-end coupled. Reproduced with permission from [55]. Copyright © 2014 by M. Sadegh Hajhashemi.

**Figure 7 micromachines-12-01361-f007:**
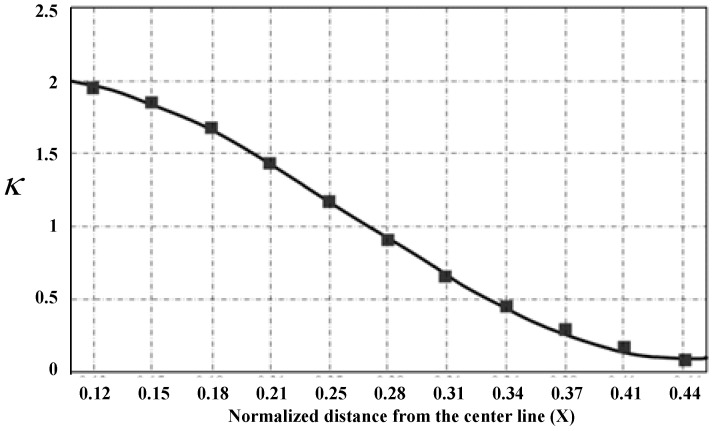
Coupling ratio of resonator varied with the location of beam. Reproduced with permission from [55]. Copyright © 2014 by M. Sadegh Hajhashemi.

**Figure 8 micromachines-12-01361-f008:**
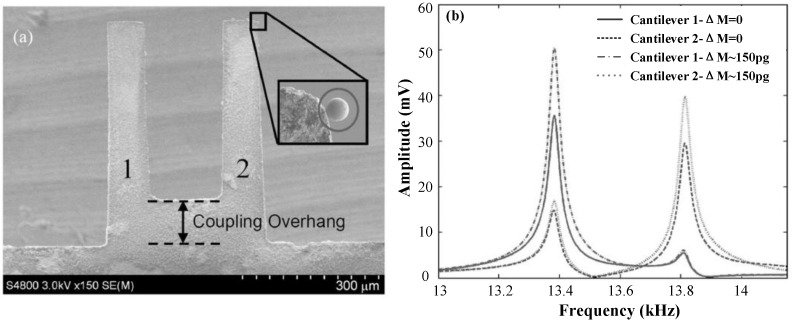
(**a**) SEM image of the coupled cantilevers resonator; (**b**) frequency change in coupled resonator before and after the induced mass. Reproduced with permission from [56]. Rights managed by AIP Publishing.

**Figure 9 micromachines-12-01361-f009:**
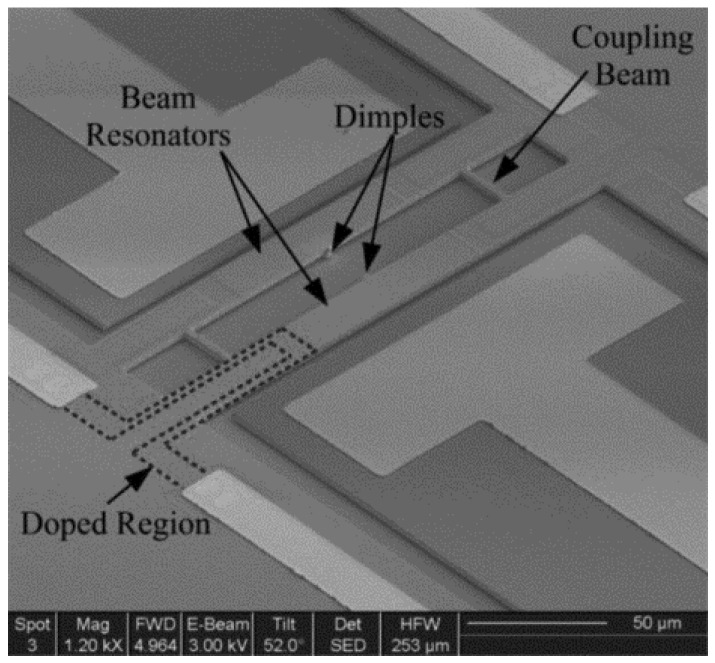
SEM image of the coupled cantilevers resonator. Reproduced with permission from [58]. Copyright © 2016, IEEE.

**Figure 10 micromachines-12-01361-f010:**
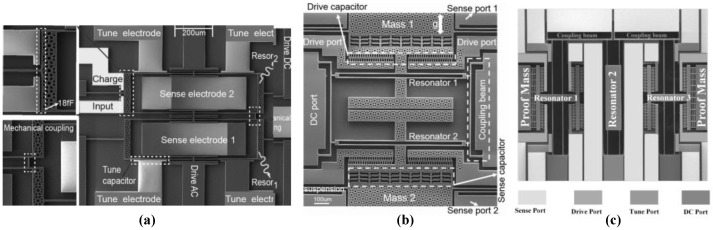
SEM image of two resonators weakly coupled ME-SCRB sensor: (**a**) electrometer; Reproduced with permission from [63]. Copyright © 2016, IEEE. (**b**) accelerometer; Reproduced with permission from [64]. Copyright © 2015, IEEE. (**c**) three resonators weakly coupled ME-SCRB sensor for acceleration measurement. Reproduced with permission from [66]. Copyright © 2018, IEEE.

**Figure 11 micromachines-12-01361-f011:**
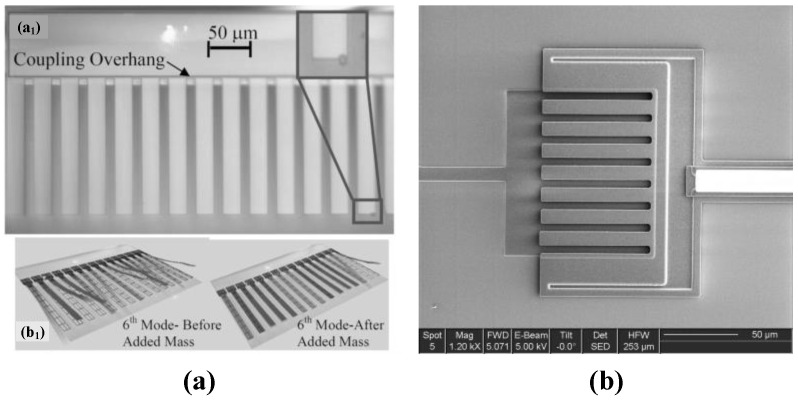
SEM photo of (**a**) weakly coupled ME-SCRB with cantilever array. Reproduced with permission from [57]. Rights managed by AIP Publishing. (**b**) Strongly coupled ME-SCRB with cantilever array. Reproduced with permission from [55]. Copyright © 2014 by M.Sadegh Hajhashemi.

**Figure 12 micromachines-12-01361-f012:**
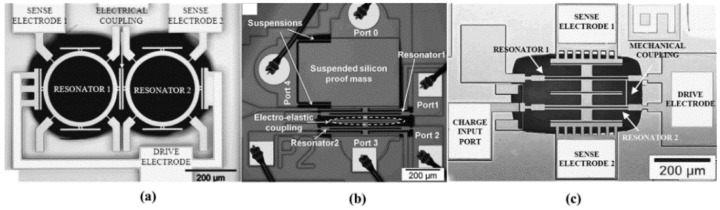
The Photos of electrically coupled resonators (**a**) mass sensor. Reproduced with permission from [77]. Rights managed by AIP Publishing. (**b**) displacement sensor. Reproduced with permission from [78]. Copyright © 2012, IEEE. (**c**) electrometer. Reproduced with permission from [79]. Copyright © 2010, IEEE.

**Figure 13 micromachines-12-01361-f013:**
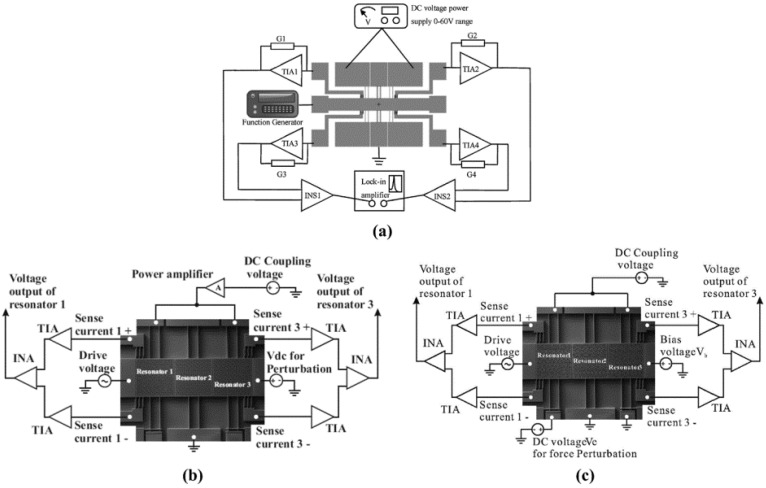
The electrically coupled three resonators (**a**) mass sensor. Reproduced with permission from [81]. © 2018 Elsevier B.V. (**b**) stiffness sensor. Reproduced with permission from [84]. Copyright © 2015, IEEE. (**c**) force sensor. Reproduced with permission from [87]. © 2015 Elsevier B.V.

**Figure 14 micromachines-12-01361-f014:**
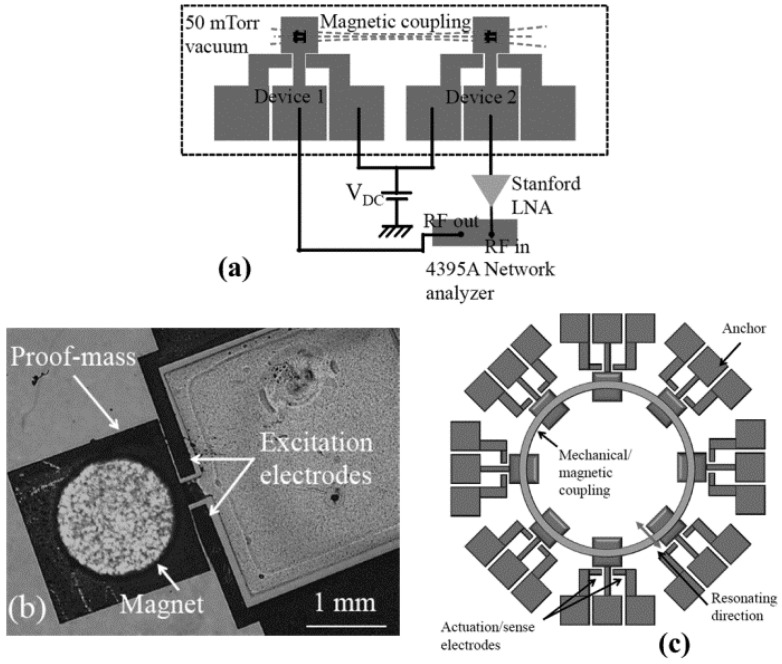
The magnetically coupled resonators: (**a**) Schematic of the experimental setup; (**b**) Optical microscope image; (**c**) A conceptual gyroscope. Reproduced with permission from [89]. Copyright © 2014, IEEE.

**Figure 15 micromachines-12-01361-f015:**
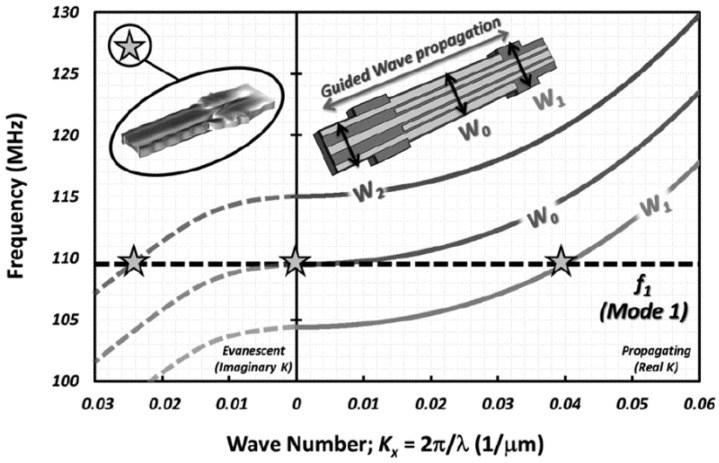
Dispersive behavior of propagating waves. Reproduced with permission from [109]. Copyright © 2013, IEEE.

**Figure 16 micromachines-12-01361-f016:**
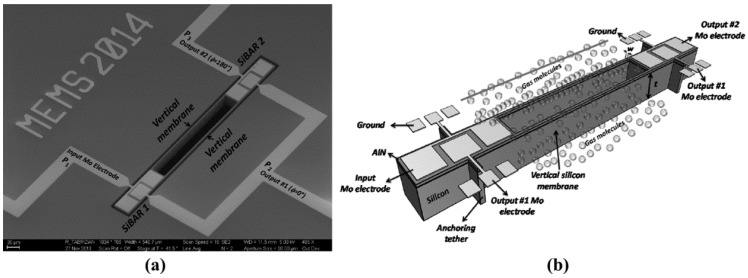
(**a**) SEM photo of device; (**b**) Schematic view of WCRB sensor for pressure measurement. Reproduced with permission from [111]. Copyright © 2014, IEEE.

**Figure 17 micromachines-12-01361-f017:**
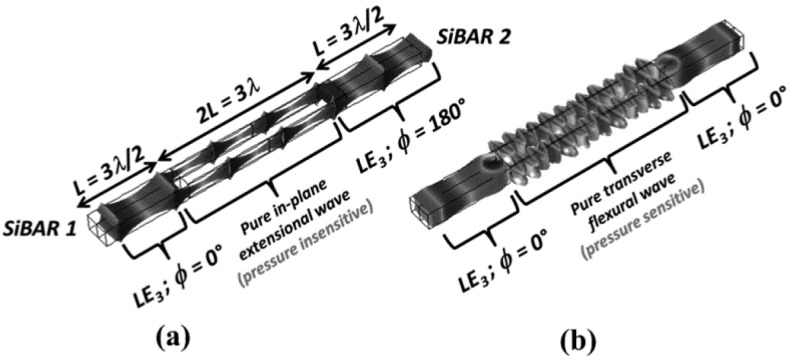
Mode shapes of (**a**) pressure insensitive extensional mode; (**b**) pressure sensitive flexural mode. Reproduced with permission from [111]. Copyright © 2014, IEEE.

**Figure 18 micromachines-12-01361-f018:**
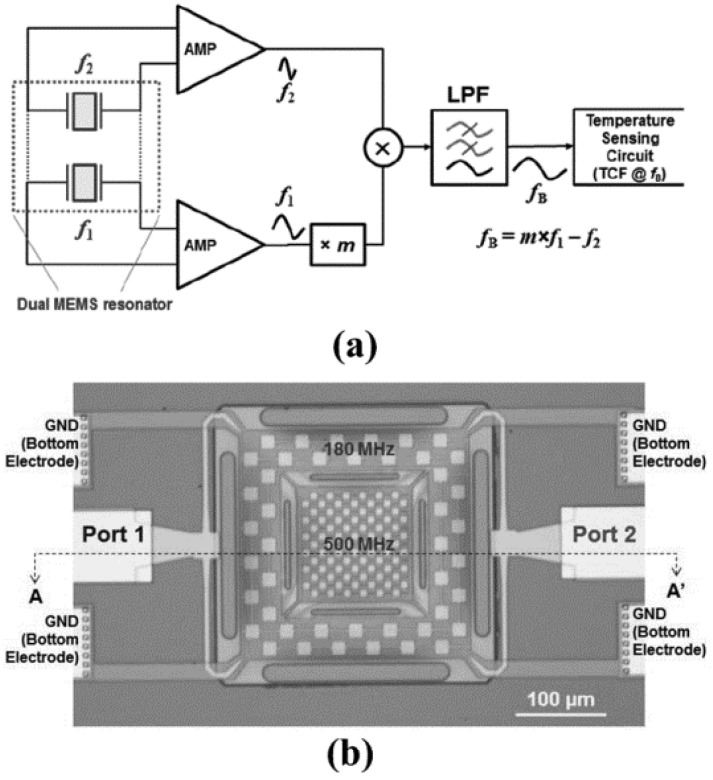
(**a**) schematic of beat frequency-based sensor system; (**b**) Optical picture of the concentric dual AlN MEMS Resonators. Reproduced with permission from [113]. Copyright © 2017, IEEE.

**Figure 19 micromachines-12-01361-f019:**
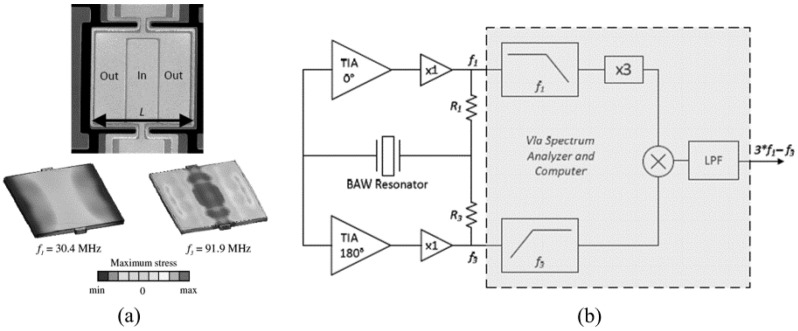
(**a**) SEM photo of device and the vibrating mode; (**b**) block diagram of Q-UCRB sensor. Reproduced with permission from [114]. Copyright © 2011, IEEE.

**Figure 20 micromachines-12-01361-f020:**
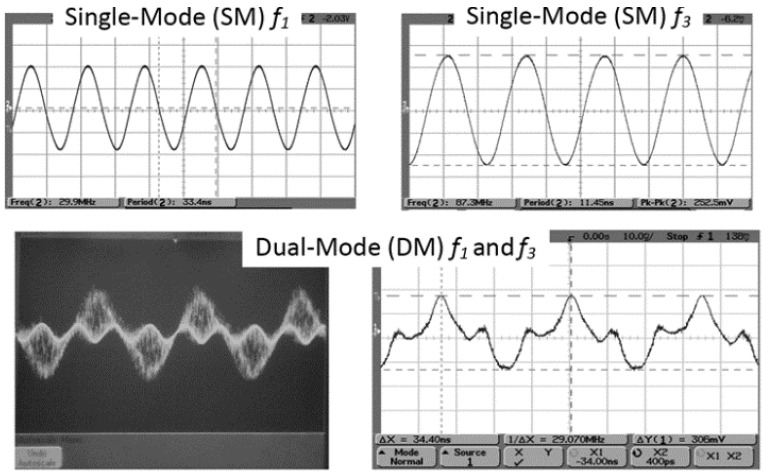
(**top**) Oscillator response for *f*_1_ and *f*_3_ single-mode excitation; (**bottom left**) dual-mode excitation at *f*_1_ and *f*_3_; (**bottom right**) dual-mode output signal after averaging. Reproduced with permission from [114]. Copyright © 2011, IEEE.

**Figure 21 micromachines-12-01361-f021:**
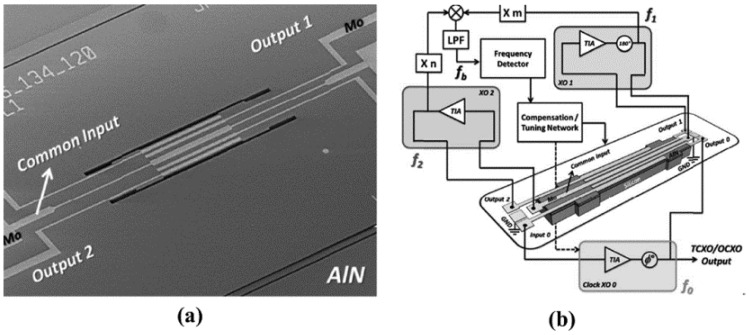
(**a**) SEM image; and (**b**) Schematic block diagram of a reference oscillator for Q-UCRB sensor for temperature sensing. Reproduced with permission from [109]. Copyright © 2013, IEEE.

**Figure 22 micromachines-12-01361-f022:**
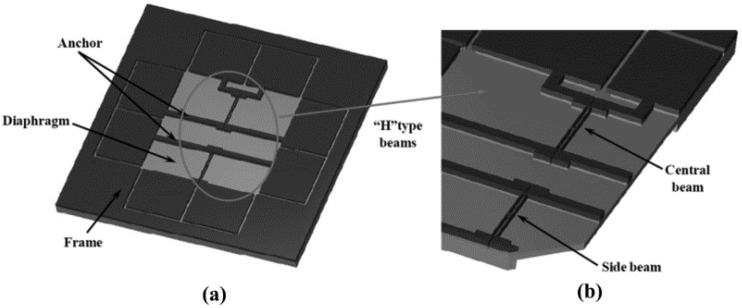
(**a**) Overall and (**b**) Partial enlarged SEM schematic of pressure sensors. Reproduced with permission from [118].

**Figure 23 micromachines-12-01361-f023:**
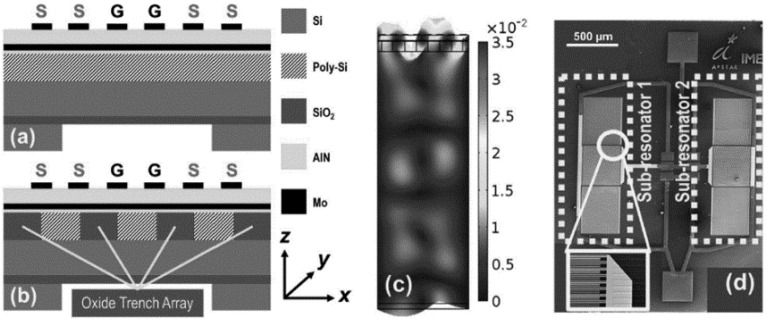
Dual-frequency resonating device (**a**) resonator without OTA; (**b**) resonator with OTA; (**c**) Displacement of the quasi-SAW mode shape of resonator; (**d**) SEM image of device. Reproduced with permission from [119]. Copyright © 2017, IEEE.

**Figure 24 micromachines-12-01361-f024:**
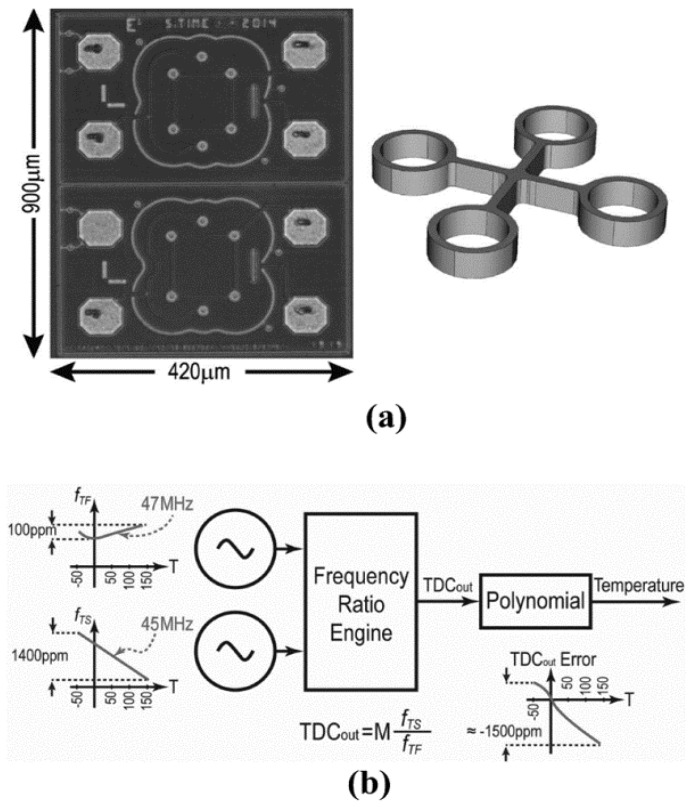
(**a**) 3-D view of MEMS resonators and the MEMS dies photo; (**b**) Architecture of a temperature sensor operating based on measuring the ratio of the frequencies. Reproduced with permission from [120]. Copyright © 2017, IEEE.

**Figure 25 micromachines-12-01361-f025:**
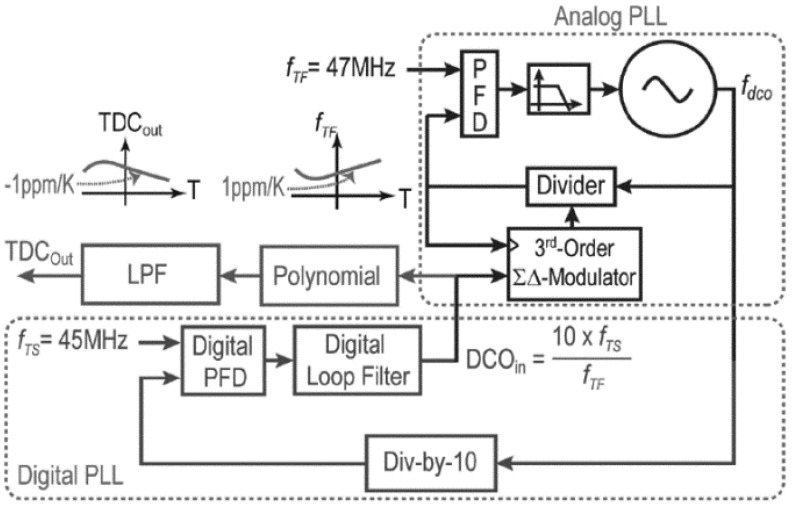
Block diagram of temperature measurement with the PS-UCRB sensor. Reproduced with permission from [120]. Copyright © 2017, IEEE.

**Table 1 micromachines-12-01361-t001:** Properties of single resonator, dual and triple resonators.

	Sensitivity of Frequency	Sensitivity of Eigenstate	Common Change Rejection
Single resonator	1/2	Non	Non
Dual resonators	Strongly-coupled	Weakly-coupled	Can
	*K*/(2*κ*)/(two to three orders) [123]	*K*/(2*κ*) [123]	
Triple resonators	Strongly-coupled	Weakly-coupled	Can
	Non	4(*K*_2_− *K*_eff_ + *K*_c_)/*K*_c_	

**Table 2 micromachines-12-01361-t002:** Comparisons of properties of different ME-SCRB sensors.

Number of Coupled Resonators	Sensor	Change of Frequency	Weakly Coupled	Strongly Coupled	Improvement
			Shift of Eigenstates	Beat Frequency	
Dual resonators	Mass sensor [56]	0.01%	5–7%	----	two orders
	Mass sensor [58]	----	----	yes	20%
	Electrometers [63]	283.56 ppm	663,751 ppm	----	three orders
	Accelerometer [64]	1035 ppm/g	312,162 ppm/g	----	302 times
	Mass sensor [68]	0.03%/pg	2.5%/pg	----	two orders
	Mass sensor [69]	−0.17	422	----	2482 times
Multiple resonators	Mass sensor [57]	0.1%	10–100%	----	two to three orders
	Mass sensor [55]	4.2 kHz	----	High coupling ratio High sensitivity	----
	Accelerometer [66]	11.46 Hz/g	705,000 ppm/g	----	1410 times

**Table 3 micromachines-12-01361-t003:** Comparisons of properties of different EL-SCRB sensors with two resonators.

Sensor	Shift of Frequency	Shift of Eigenstates	Improvement
Mass sensor [77]	0.00237%	4.32% or 3.448%	Two orders
Displacement sensor [78]	maximum 0.005%	maximum 1.8%	Three orders
Electrometer [79]	maximum 0.006%	maximum 1.8%	Nearly three orders
Mass sensor [80]	0.02%	221%	More than four orders

**Table 4 micromachines-12-01361-t004:** Comparisons of EL-SCRB sensors for stiffness and force detections.

Sensor Type	Reference	Output of Sensor	Amplitude Ratio	Improvement	Improvement
				As Frequency Shift	As Eigenstate Shift
Mass sensor	[81]	Amplitude ratio	25.31%	Nearly two orders	Nearly two times
	[82]	Amplitude ratio	0.4%	Nearly two orders	----
	[83]	Amplitude ratio	35.6	More than three orders	----
Stiffness sensor	[84]	Amplitude ratio	13,558	More than three orders	56 times
	[79]	Shift of eigenstates	275	Two orders	----
	[133]	Shift of frequency	0.5	----	----
Force sensor	[87]	Amplitude ratio	4.9e6/N	Two orders	More than three orders
	[79]	Shift of eigenstates	1478/N	----	----
	[134]	Shift of frequency	8995/N	----	----

**Table 5 micromachines-12-01361-t005:** Comparison of resolution limit model of 2-DoF WCRs.

Reference	Resolution Limit Model
J. Juillard [96,97]	RAR=2NQFΔf12
A. Seshia [98]	$R_{\mathrm{AR}}\approx 8\kappa \sqrt{\frac{E_{\mathrm{th}}\Delta f}{2E_{c}Q\omega_{r}^{\mathrm{eff}}}}$
Chang [99]	RAR≈22κNFΔf12=22κ4κBTcΔfF

**Table 6 micromachines-12-01361-t006:** Comparison of resolution limit model using frequency and AR output metrics. Adapted from [99].

Output		Resolution Limit Model
Frequency Output Metric		1QNFΔf12
AR Output Metric	2-DoF	22κNFΔf12
	3-DoF	22κ2a−1NFΔf12
	4-DoF	22κ3(a−1)2NFΔf12

**Table 7 micromachines-12-01361-t007:** Comparisons of properties of WCRB sensors.

Reference	Coupling Way	Sensitivity	Merits
[109]	3rd width-extensional mode (WE3) and cross-sectional distortional mode	−8300 ppm/°C	High temperature sensitivity
[110]	1st width-extensional mode (WE1) and 2nd Width-shear mode (WS2)	----	Q enhancement and TCF reduction
[111]	3rd length-extensional mode (LE3) and transverse flexural mode	346 ppm/kPa	High pressure sensitivity

**Table 8 micromachines-12-01361-t008:** Comparisons of Q-UCRB sensors.

Sensor	Mode	Output
Temperature sensor [114]	Fundamental (*f*_1_) and third-order (*f*_3_) length extensional mode	TCF 162 ppm/°C
Temperature sensor [115]	In-plane width-shear (WS) and width-extensional (WE) modes	TCF 1480 ppm/°C
Temperature sensor [109]	in-plane and out-of-plane lamb wave mode	TCF 8292 ppm/°C
Temperature and mass-loading sensor [116]	FBAR	Mass loading and the temperature
Pressure sensor [117]	FBAR	Mode 1: 1.642 ppm(kPa)^−1^ Mode 2: 0.1764 ppm(kPa)^−1^

**Table 9 micromachines-12-01361-t009:** Comparisons of PS-UCRB sensor.

Sensor	Temperature Compensation Method	Output
Pressure sensor [118]	Algorithm compensation	Low error less than ±0.01%
High pressure sensor [119]	Algorithm compensationand material matching	Non-linearity of 2.28 % F.S
Temperature sensor [120]	Hardware deducing	High resolution of 20 μK

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
