# Peer review of "Dual-Resonator-Based (DRB) and Multiple-Resonator-Based (MRB) MEMS Sensors: A Review"

_micromachines, 2021, doi:10.3390/mi12111361_

Round 1
Reviewer 1 Report
The article is a review article dealing with MEMS sensors operating with two or multiple resonators. The authors have undertaken a work of classification according to the type of coupling between the resonators of the system and for each class of coupling, they have detailed the different ways to achieve the type of coupling in question. On this point, there is a real work to compile information in quite diverse areas, in terms of applications or operating principles, although all falling under MEMS sensors using resonators.
Nevertheless, in my opinion, the overall scientific interest of the work done is not obvious for several reasons.
First of all, even if the synthesis work has been underlined, it appears that the bibliographical references used are for many old and it would be necessary to update these references, given that a consequent number of publications exist on the last two years, on devices using mode localization for example.
The state of the art on the WCRB and UCRB parts allows the authors to conclude that these devices show a significant sensitivity without this assertion being really supported by a comparison with other devices or other technologies.
The two major arguments put forward to prove the interest of dual or multiple coupled resonators are a high normalized sensitivity compared to a single resonator. However, this argument must be balanced by the fact that we are comparing two normalized sensitivities based on very different physical measurements. In the case of simple resonators, a frequency is measured whereas in the case of coupled resonators, a vibration amplitude is measured. The question that arises is the comparison of the resolution of each of the two types of measurement. If the normalized sensitivity is two or three orders of magnitude greater for the displacement measurement, what about the resolution compared to that of a frequency measurement?
The second argument is that of common mode rejection. The authors propose only one reference on this point, which dates from 2009. This argument should therefore be supported by a more thorough bibliographic study. The following review article may be useful: Vinayak Pachkawade, "State-of-the-Art in Mode-Localized MEMS Coupled Resonant Sensors: A Comprehensive Review", DOI: 10.1109/JSEN.2021.3051240.
Reviewer 2 Report
Authors reviewed the dual-resonator-based (DRB) and multiple-resonator-based (MRB) MEMS Sensors. Authors compared single resonator and coupled resonators, and then analyzed different DRB and MRB sensors thoroughly, including SCRB, WCRB, UCRB sensors and their branches. Finally, the future prospects are described. However, there are many unclear explanations in this manuscript. My comments are addressed as follows.
- The introduction should provide more detail of the background. For example, the comparison of resonator-based MEMS sensors and other sensors.
- The equation (1) is wrong and other equations as well should be double checked.
- Some figures should be enlarged to be clearly identify, such as Figure 8.
- The figures of the same device type are suggested to be combined into one figure as possible for direct reviewing. For example, there are 4 individual figures introducing ME-SCRB, i.e., Fig.11-14.
- Every variation in the equations should be clearly described, such as x and X in equation (18).
- The descriptions in the tables should be unified and precise, for example, “1000 times” and “nearly three orders” in Table 3 are the same.
- Language errors should be removed, for example, in the line 373 “From the point of view fabrication”.
- For a review paper, some descriptions of the same MEMS design should be discussed in detail, e.g., line 561-583. Such descriptions can be easily found in the references for readers and the authors just need to propose the main points.
- In fourth part, the session of conclusions and future Perspectives is suggested to illustrate individually.
Round 2
Reviewer 2 Report
This manuscript is well-organized and good quality. I recommend it can be published.